# Post-Mortem Interval of Human Skeletal Remains Estimated with Handheld NIR Spectrometry

**DOI:** 10.3390/biology11071020

**Published:** 2022-07-06

**Authors:** Verena Maria Schmidt, Philipp Zelger, Claudia Wöss, Christian Wolfgang Huck, Rohit Arora, Etienne Bechtel, Andreas Stahl, Andrea Brunner, Bettina Zelger, Michael Schirmer, Walter Rabl, Johannes Dominikus Pallua

**Affiliations:** 1Institute of Forensic Medicine, Medical University of Innsbruck, Muellerstraße 44, 6020 Innsbruck, Austria; verena-maria.schmidt@i-med.ac.at (V.M.S.); claudia.woess@i-med.ac.at (C.W.); walter.rabl@i-med.ac.at (W.R.); 2University Clinic for Hearing, Voice and Speech Disorders, Medical University of Innsbruck, Anichstrasse 35, 6020 Innsbruck, Austria; philipp.zelger@tirol-kliniken.at; 3Institute of Analytical Chemistry and Radiochemistry, University of Innsbruck, 6020 Innsbruck, Austria; christian.w.huck@uibk.ac.at; 4University Hospital for Orthopaedics and Traumatology, Medical University of Innsbruck, Anichstraße 35, 6020 Innsbruck, Austria; rohit.arora@tirol-kliniken.at; 5Spectral Engines GmbH, Weißkirchener Straße 2-6, 61449 Steinbach, Germany; etienne.bechtel@spectralengines.com (E.B.); andreas.stahl@spectralengines.com (A.S.); 6Institute of Pathology, Neuropathology, and Molecular Pathology, Medical University of Innsbruck, Muellerstrasse 44, 6020 Innsbruck, Austria; andrea.brunner@i-med.ac.at (A.B.); bettina.zelger@i-med.ac.at (B.Z.); 7Department of Internal Medicine, Clinic II, Medical University of Innsbruck, Anichstrasse 35, 6020 Innsbruck, Austria; schirmer.michael@icloud.com

**Keywords:** post-mortem interval, NIR spectrometry, handheld tool, deep learning

## Abstract

**Simple Summary:**

Handheld NIR spectrometry represents a non-destructive method to estimate post-mortem interval with a short sample-preparation time. Based on a deep-learning technique for post-mortem interval approach (ranging from 1 day up to 2000 years) an estimation of post-mortem interval with an accuracy of almost 100% in bones is possible.

**Abstract:**

Estimating the post-mortem interval (PMI) of human skeletal remains is a critical issue of forensic analysis, with important limitations such as sample preparation and practicability. In this work, NIR spectroscopy (NIRONE^®^ Sensor X; Spectral Engines, 61449, Germany) was applied to estimate the PMI of 104 human bone samples between 1 day and 2000 years. Reflectance data were repeatedly collected from eight independent spectrometers between 1950 and 1550 nm with a spectral resolution of 14 nm and a step size of 2 nm, each from the external and internal bone. An Artificial Neural Network was used to analyze the 66,560 distinct diagnostic spectra, and clearly distinguished between forensic and archaeological bone material: the classification accuracies for PMIs of 0–2 weeks, 2 weeks–6 months, 6 months–1 year, 1 year–10 years, and >100 years were 0.90, 0.94, 0.94, 0.93, and 1.00, respectively. PMI of archaeological bones could be determined with an accuracy of 100%, demonstrating the adequate predictive performance of the model. Applying a handheld NIR spectrometer to estimate the PMI of human skeletal remains is rapid and extends the repertoire of forensic analyses as a distinct, novel approach.

## 1. Introduction

Accurate and rapid identification of human remains often depends on the post-mortem interval (PMI). The PMI must be rapidly determined to identify whether skeletal human remains are of archaeological interest or forensically relevant. Especially recent skeletal remains not older than a few decades are of significant interest to law enforcement [1,2,3]. Depending on the legal system of each country, the time it takes for a case to be relevant varies, for example, with 15 years in Portugal and 50 years in Germany [4,5,6,7]. In skeletal remains, the PMI can be difficult to estimate following the decomposition of soft tissues.

Despite considerable research, there is still no definitive method for estimating the PMI of skeletal remains [8]. The difficulties arise from the complex degradation processes, referring to chemical, physical, and biological processes that occur between bones and the environment where they are located [4]. The type of degradation is not well understood [9], but different parts of the bone have different characteristics and chemical compositions. The organic phase of the bone consists of collagen, lipids, and proteins, which are found within the medullary part. The cortical part of the bone contains organic elements and minerals, including magnesium and hydroxyapatite crystals. Therefore, a portion of the cortex is inorganic and surrounded by the organic phase [10,11]. Degradation is affected by both internal and external factors [4,10]. Internal factors include age, diseases, and diet [10]. External factors include the environment and the characteristics of the soil where the body is buried.

The PMI is typically determined from skeletal remains using morphological analysis, a time-related degradation estimation of other discoveries such as clothing and personal items, or chemical and physical examination [12,13]. Despite its less objective nature, this method involves measuring the external changes in bones [14]. Observing the presence or absence of ligamentous structures and the rate at which fats and other organic matter leaches out, the bones themselves can explain the time since death [15]. The forensic field has proliferated as a branch of analytical chemistry [9]. Several studies focus on bones’ chemical and physical changes after death and the varied factors affecting the diagenetic process [16,17,18]. So far, techniques applied to determine the time elapsed since death of skeletal remains include histological examination [19,20,21], reaction with a mineral acid, reaction with benzidine, nitrogen loss [22], proteomics [23,24,25,26], metabarcoding [27,28], degradation of lipids, remnants of fat-transgression [29], high performance liquid chromatography–tandem mass spectrometry [30], UV-Vis spectroscopic methods [11,31,32,33,34], radioisotope measurements [27,35,36,37], luminol chemiluminescent reaction [3,36,38,39,40], X-ray diffraction [10,41,42], micro-computed tomography [42], and infrared (IR) spectroscopy [4,8,9,33,42,43,44,45,46,47,48,49,50,51,52]. Some of these methods are destructive, others lack objectivity.

Therefore, objective methods such as near-infrared (NIR) spectroscopy have the potential for more rapid and accurate estimations of PMI. NIR spectroscopy is optical absorption spectroscopy situated between the visible and mid-infrared region, covering a wavelength range between 780 and 2500 nm. NIR spectra result from fundamental vibrations and overtones of hydrogen bonds such as O-H, C-H, and N-H, enabling quick analysis for a wide range of materials with many different physicochemical parameters and the chemical composition. Many physicochemical parameters can be analyzed quantitatively using NIR spectroscopy, such as particle size, hardness, dissolution rate, compaction force, and water content [53,54,55]. It is a rapid technique for recording a spectrum within only a few seconds. A NIR spectrometer consists of the radiation source, wavelength selector, interferometer, and detector, interfaced with optics, which are available both as benchtop spectrometers for use in laboratories and as autonomous spectrometers used on-site.

NIR spectroscopy’s physical principles make it suitable for miniaturization, and NIR spectrometers have achieved outstanding advances when available in compact technology. Commonly accepted classification of the deployability of the instrumentation distinguishes the transportable, ‘suitcase-type’ and handheld spectrometers. Using state-of-the-art handheld spectrometers for PMI research might provide a valuable extension to conventional ways. In the past decade, the emergence of handheld spectrometers marked a significant turning point in the evolution of the practical applications of NIR spectroscopy. The most recent years have now led to an emerging new class of spectrometers, which are miniaturized, reaching a weight lower than 100 g, with ultra-miniaturized sensors that are compact enough to be built directly into a smartphone device. The miniaturization of sensors has enabled a broader array of NIR spectroscopy applications [56].

Still, several concerns have been raised regarding miniaturized spectrometers, as the new technologies incorporated into handheld devices are much less uniform. The compact technologies show different performances, sometimes with narrower spectral regions or lower resolution. Therefore, data are needed for specific demands. This work focuses on a thorough systematic evaluation of the PMI estimation of human skeletal remains with a handheld spectrometer for NIR spectroscopy. The aim is to assess the potential value of these handheld NIR spectrometers for estimating the PMI.

## 2. Materials and Methods

### 2.1. Sample Collection and Ethical Considerations

Recent forensic bone samples (*n* = 99) were routinely collected for molecular genetic identification purposes during an autopsy at the University Institute of Forensic Medicine, and archaeological bone samples from medieval times were collected from European excavation sites (*n* = 5). The bone samples with 0–2 weeks PMI (class 1, *n* = 32), 2 weeks–6 months PMI (class 2, *n* = 46), 6 months–1 year PMI (class 3, *n* = 11), 1 year–10 years PMI (class 4, *n* = 10), and >100 years PMI (class 5, *n* = 5) were obtained from 16 female and 88 male human remains. The class intervals were based and adjusted according to the time frames used in the §57 of the Austrian criminal code [57]. Classification of PMI was based on investigations by the police and forensic needs before NIR spectrometry. In the case of uncertain conventional estimation of PMIs, the average result was used for classification. Typical anthropological methods (e.g., pelvic bone, skull characteristics [12,58]) and DNA typing were routinely assessed for sex estimation (data not shown). The diaphysis of the femur of forensic and archaeological bone samples was used for analyses. Using a hand saw, one half transversal section was cut from each bone with a thickness of about 7 mm. The cutting planes were cleaned from periost and bone marrow and dried for a few days at room temperature. NIR spectrometry was applied for this study before additional forensic analyses.

The study was conducted according to the ICH-GCP guidelines and the declaration of Helsinki. Ethical approval was obtained from the local ethics commission (EK: 1357/2021).

### 2.2. Measurement System

The measurements were performed using a “LabScan” setup of eight devices with NIR detectors (NIRONE^®^ Sensor S2.0; Spectral Engines, 61449 Steinbach, Germany), minimizing variations due to the technical setting. The eight sensors are placed in mechanical holders in this setup and linked via USB hubs to a computer for data storage.

The measurements of the NIR devices are based on a relative calibration process, which is different from the classical benchtop devices, evaluating each sensor’s “response function” along the wavelength axis. Before using the NIR devices, reference material is measured, which acts as an optimal reflector and provides access to the specific response function of each sensor in its wavelength regime.

Overall, each sample was measured 40 times in the range of 1950 to 1550 nm wavelength. After these 40 measurements, each sample was moved to the next detector, as shown in Figure 1. This step was repeated another seven times to assess the statistical fluctuations from the hardware production and the measurement process itself. A second round was then performed for the internal bones. Thus, optimal parameter space coverage was achieved for the deep-learning approach. A dataset of 66,560 measurements was created.

### 2.3. Data Preparation

Since the individual devices are not calibrated to produce a transferable signal output, the spectral data from each instrument is divided by its reference measurement. The result describes the reflectance of the respective sample. Additionally, the design of the devices does not ensure a reproducible signal height due to potential differences in the sample placement. Therefore, the data needed to be normalized, which was done with a “Standard Normal Variate” (SNV) to evaluate the relative differences along the wavelength axis, making measurements between different devices comparable.

In the last step, the data are filtered by multiple algorithms to ensure high-quality data. Measurement errors, such as the movement of the device while measuring, can produce recognizable artifacts that can be removed automatically with a very high level of confidence.

### 2.4. Model Creation

An Artificial Neural Network (ANN) was chosen to evaluate the spectral data. Neural networks, especially deep neural networks, can learn complex correlations in data sets and outperform traditional machine learning approaches on various tasks. In this work, the input data consists of an n-dimensional data vector that represents the reflection spectra recorded by the NIR devices. The network consists of several fully connected layers intercepted by dropout layers and regularization layers to prevent overfitting. A controlled data split was performed to avoid cross-correlations between training and test datasets and to avoid biases in the evaluation metrics. Slight label weight adjustments were used to address the asymmetry between output classes and reduce prediction biases correlated to over/undersampling of certain age classes due to sample availability.

In Figure 2, a schematic illustration of the workflow is presented. As an initial step of the given workflow, a recording and preprocessing step was implemented by measuring reference spectra, spectra, and calibration. The next step is to divide the dataset into their respective age classes. The labeled spectra (according to the age classes) are then divided into a training and test set according to the age classes. The training set is used to train and refine the ANN. The test set is used to evaluate the predictive abilities of the ANN.

## 3. Results

Measured human skeletal remains are summarized in Table 1.

### 3.1. Optimization of the Measurement Location

Optimizing the measurement location of a sample for NIR experiments is essential to obtaining high-quality results. The model proved to be invariant to the sample spot and robust enough to be applied to all spots and sides of the bone sample. For this purpose, all samples were measured from the external and internal bone. The effects of the measurement location on spectra are illustrated in Figure 3. It can be demonstrated that there is no detrimental effect of the beam focus settings on the spectral shape. In total, 66,560 measurements from the external and internal bones were analyzed. The results show that the spectra taken from the internal and external samples cannot be distinguished by their respective spectra.

### 3.2. Classification of Post-Mortem Interval

The spectra obtained by NIR spectrometry provide a high dimension of chemical information, demonstrating the sample’s molecular fingerprint and observing different structures or mechanisms linked to the PMI. Figure 4A shows the mean spectra of the five age classes, revealing PMI class-specific profiles from 1950 to 1550 nm. Prominent spectral features are peaks at 1800 and 1600 nm.

### 3.3. Confusion Matrix of Classification Results

Figure 4B shows the classification result as a confusion matrix. The diagonal elements of such a confusion matrix represent the percentage of correctly classified elements. The bright diagonal elements of the confusion matrix represent the percentage of correctly classified elements, with an accuracy ranging between 100% for the archeological samples (fifth class) and 90% for samples with a PMI between 0 and 2 weeks (first class). In total, 13 elements were classified with a PMI of “some weeks”, which allows an assignment to two classes. Unfortunately, all information was not available in the set of samples analyzed. Samples were included with often undefined PMI descriptions (9 samples with PMI of uncertain number of less than 6 weeks, and 2 samples with uncertain PMI of less than 10 years), different find spots (e.g., forest, flat, buried, water), and thermal alterations (plane crash = 3 and apartment fire = 1).

### 3.4. Practicability of Handheld NIR Spectrometry

NIR measures the composition of a sample based on the absorption of infrared radiation by vibrational transitions in covalent bonds. This method represents a rapid and non-contact technique without the requirement of additional sample preparation or extractions. The combination of 8 handheld devices can be considered a small desktop device. However, each single device combined with ANN software developed according to this study can be used as a handheld device, which can be used independently from a desk. Using the handheld NIR spectrometer and a mobile software application takes only seconds to estimate the PMI of human skeletal remains (Figure 5). 

## 4. Discussion

There is a need for forensic anthropologists and pathologists to continue investigating precise methods for PMI estimation of skeletal remains [56]. This study aimed to evaluate the suitability of handheld NIR spectrometry to produce distinct and diagnostic spectra distinguishing between forensic and archaeological bone material. Therefore, handheld NIR spectrometry measurements and a deep-learning approach to correlate the PMI with the detected spectral properties were used. Optimization of the measurement location presented no difference between the external and internal bones. The main focus of this study was the training of an ANN that allows for the estimation of the PMI of human bones with an NIR Handheld Spectrometer. An exact assignment to chemical compounds cannot be made because the incoming infrared light excites every state that matches its energy. Such states can be a single atom or molecule energy levels, vibrational or rotational states, but also vibrational or rotational states of complex molecules or lattice vibrations in general. The presented smooth spectrum, without any distinct peak leads, does not allow for the extraction of distinct molecule spectra. The spectra between the first four and the fifth class vary to such an extent in amplitude and shape, as shown in Figure 4A, that the distinction between the first four and the fifth class could even be made without a machine learning approach. This indicates that the limited number of samples did not impose a limitation on the results of this study. Additionally, the gender imbalance between the samples most likely does not lead to a bias in the results. This observation is also demonstrated in [59], that the inter-gender variation is much bigger than the gender difference for most bone-related parameters.

As shown in a confusion matrix (Figure 5), ANN classification results clearly distinguish between forensic and archaeological samples. The machine learning approach allows for the extraction of intrinsic information from spectra without knowing what information one is looking for. The neural network learns the features of the data and extracts the required information to form a decision on the age classes. The learned feature extraction capabilities of the network are encoded in the weights of the network. The neural network forms an abstract representation of the data. Those weights and the derived representations do not present themselves in terms of wavenumbers or similar terms. The result shows a perfect classification of samples for the fifth class (archaeological samples) and the lowest accuracy (90%) for the first class. The age uncertainty of the first-class results in a blurring of the class boundary between the first and second classes, e.g., a bone with a determined age of 14 (±1) days puts the dataset in both the first and the second classes. The same problem can be found analogously in the first temporal elements of the second class. There are some elements classified with “some weeks”, which, within the error, allows an assignment to both classes. This also explains the lowest but still above 90% accuracy in the classification of this first class. A limitation of this study is that samples with a PMI between 10 and 100 years were not available. The classification results might be explained by the differences in the chemical composition of bones. Differences in degradation and environmental effects might also explain the observed differences between the bones with different PMI. The classification results suggest that differences in the chemical composition of bones are responsible for the observed differences in NIR spectra. 

The positive results of this study indicate that this method is accurate in differentiating between skeletal remains with a PMI of less than 10 years and historical samples older than 100 years. This differentiation is crucial for forensic medicine, as further operating procedures change dramatically if the skeletal remain is a historical sample without forensic interest. The non-invasive nature of this analysis ensures the sample’s integrity before other analyses. Traditional methodologies are less objective, more expensive and time-consuming, and require specialized operators and instrumentation [60]. Overall, the immediate environment of skeletal remains induces specific degradation processes related to the PMI. NIR spectrometry appears to objectify the results of these degradation processes. Further research into specific types of environments with well-defined PMIs will be necessary to improve NIR spectrometry’s accuracy further. Comparing these data with data from vibrational spectroscopic analyses of bones, taxonomic identification, preservation mechanisms, diagenetic and thermal alteration pathways, and chemical composition [61] will improve our understating of PMI, especially during the first days.

## 5. Conclusions

This study demonstrates the potential of NIR spectroscopy coupled with a deep-learning approach as a means of rapid PMI classification of human bone samples. This approach can distinguish between forensic and archaeological human bone samples. The study highlighted the potential usefulness of the technique as a new, portable technique as more accessible instruments appear on the market. Further optimization of this technique could lead to significant advances in the area. Potential applications include distinguishing between human and animal bone fragments, on complete bones, or even on bones still in situ.

## Figures and Tables

**Figure 1 biology-11-01020-f001:**
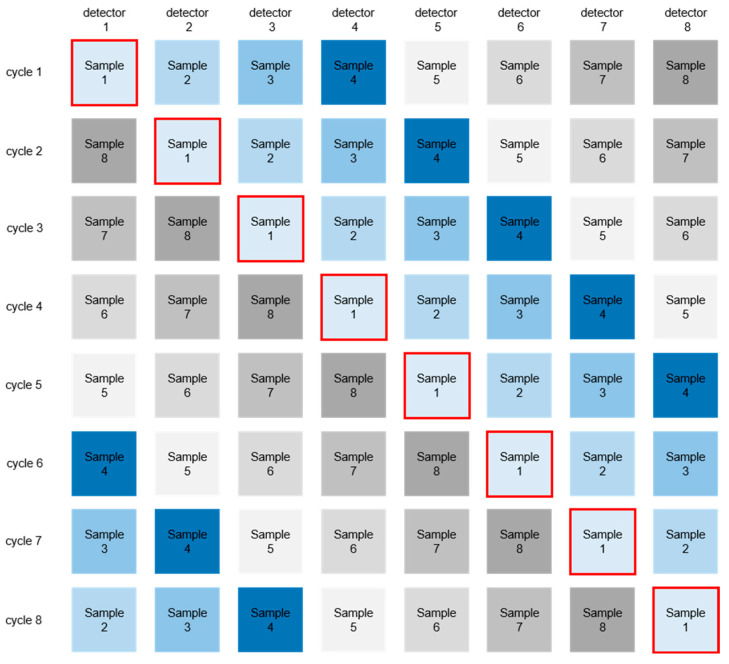
LabScan system of eight NIR detectors. Each cycle includes 40 spectra. Each sample was measured with each detector. A total of 66 560 measurements were obtained.

**Figure 2 biology-11-01020-f002:**
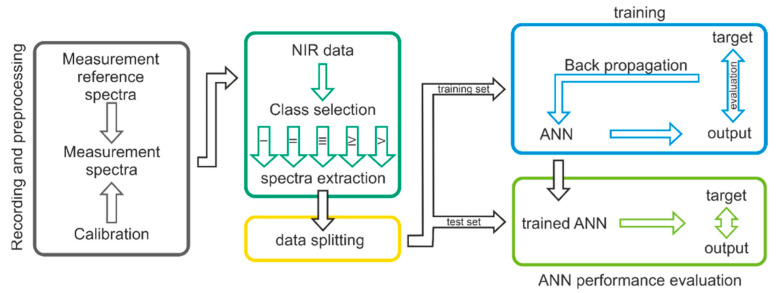
Schematic representation of the study approach: the procedure was based on extracting hyperspectral NIR information by utilizing a deep-learning-based model applying ANN.

**Figure 3 biology-11-01020-f003:**
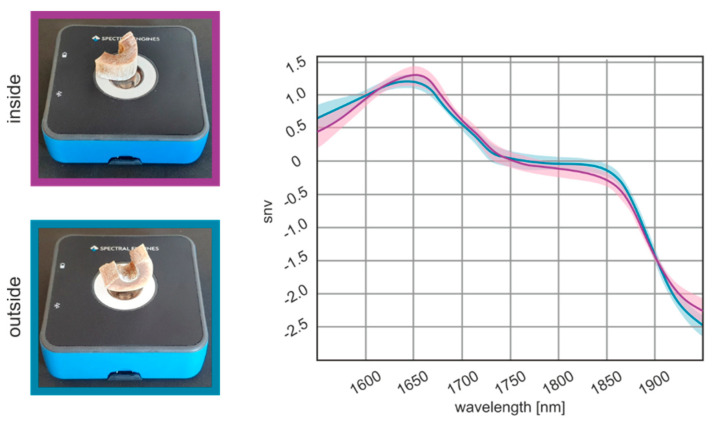
Effects of measurement location on spectra quality. NIR spectra from human bone samples measured externally and the internally, displayed in the range from 1950 to 1550 nm.

**Figure 4 biology-11-01020-f004:**
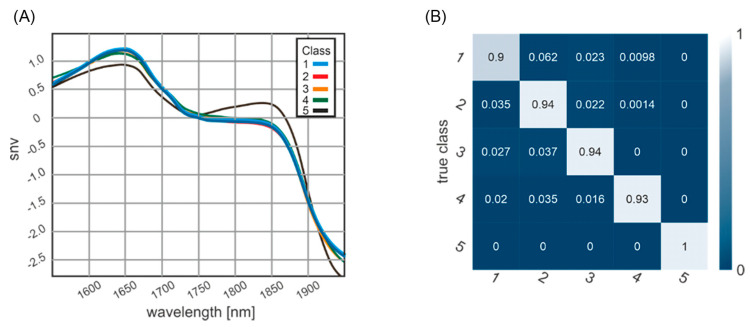
(**A**) Mean spectra are shown for the five different age classes: class 1 with PMI of 0–2 weeks (blue line; *n* = 32), class 2 with PMI of 2 weeks–6 months (red line; *n* = 46), class 3 with PMI of 6 months–1 year (orange line; *n* = 11), class 4 with PMI of 1–10 years (green line; *n* = 10), and class 5 with PMI of >100 years (black line; *n* = 5). (**B**) The confusion matrix of ANN classification shows classification accuracies between 0.90 and 1.

**Figure 5 biology-11-01020-f005:**
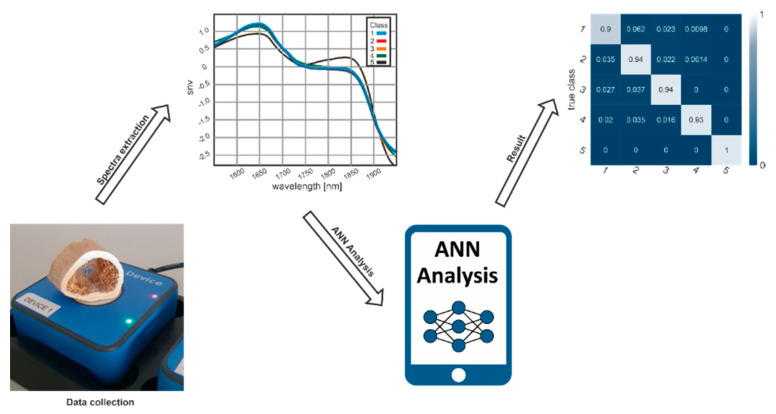
Workflow for NIR-based analysis of spectra. After a single data collection with the extraction of typical spectra, the spectra are analyzed by the ANN software to assign the predicted age class to the input data with high accuracy of almost 100%.

**Table 1 biology-11-01020-t001:** Anthropological properties and discovery of the measured human skeletal remains.

PMI	0–2 Weeks	2 Weeks–6 Months	6 Months–1 Year	1 Year–10 Years	>100 Years
flat	25	34	5	0	0
flat bathtub	1	0	0	0	0
plane crash	4	0	0	0	0
drowned	2	8	1	1	0
forest	0	3	3	7	0
forest hut	0	0	0	1	0
mountain	0	1	1	1	0
soil	0	0	0	0	1
Σ	32	46	11	10	5
median age	61	59	52.9	49	n.a
mean age	58.44	61.98	57.5	44.28	n.a
STD age	16.95	13.81	20.74	10.70	n.a
not identified age	0	0	1	3	n.a
female	3	9	2	1	1
male	29	39	9	9	2
not identified sex	0	0	0	0	2

## Data Availability

Not applicable.

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
