# Peer review of "Post-Mortem Interval of Human Skeletal Remains Estimated with Handheld NIR Spectrometry"

_biology, 2022, doi:10.3390/biology11071020_

Round 1
Reviewer 1 Report
The article is well written, with nice figures, and introduces a lot of topics very concisely. The research is novel and would be of great interest in forensic science. There are a few areas that require clarification (noted below). Further, the results need to be explicitly provided as the data appears to be largely missing.
2. Materials and Methods, 2.1 Sample Collection and ethical considerations:
· A reference is needed for where these classes of PMI come from (i.e., classes 1-5)
· ‘sex determination’ should be ‘sex estimation’ if from anthropological assessment
· ‘thigh bone’ should be 'femur'
· Please clarify what bones were used, was this all femurs or a mixture? The archaeological bones in particular are not specified.
General comments:
· There is considerable male bias in the population/data used (16 female and 88 male) – please could the authors address the potential implications of this in the discussion
· Authors frequently note the method as being non-destructive. However, the removal of bone samples from a bone/body is an invasive process and somewhat destructive to the bone. Please could this be acknowledged in the manuscript. Moreover, if the NIR could actually be used on complete bones, or even on bones still in situ (exposed but in a body) then this could also be worth noting.
· The terms ‘outside’ and ‘inside’, should be replaced with ‘external’ and ‘internal’ bone.
· Only 5 archaeological bones were used (please discuss this limitation)
· The actual PMI of the archaeological bones has not been provided, except a value of ‘up to 2000 years’ is mentioned in summary – please include full sample details in the method
· Results are largely missing – please include your data or at least an exemplar
· If known, please detail what molecules the NIR-spectrometry is/could be measuring
· It is unclear to me what exactly is indicating the age classes? Is this different peaks? Please demonstrate with data/examples.
· Regarding section 2.5 – testing of unknown samples is mentioned, it is unclear to me if the results from this are reported. What were these used for?
· The title explicitly refers to a ‘handheld’ device, however, in Figure 5, the device appears to be a small desktop device, not one that is held in a hand. I would suggest the authors reconsider the use of ’handheld’.
Author Response
Dear Reviewer,
Thank you for considering this paper and for the constructive comments. We changed the paper regarding the comments . In this response letter, we will document all our answers to the reviewers and state where the changes were applied in the final manuscript:
Reviewer 1:
The article is well written, with nice figures, and introduces a lot of topics very concisely. The research is novel and would be of great interest in forensic science. There are a few areas that require clarification (noted below). Further, the results need to be explicitly provided as the data appears to be largely missing.
- Materials and Methods, 2.1 Sample Collection and ethical considerations:
- A reference is needed for where these classes of PMI come from (i.e., classes 1-5)
AW: We added the following sentence and the reference of the Austran criminal code: The class intervals were based and adjusted according to the time frames used in the § 57 of the Austrian criminal code.
- ‘sex determination’ should be ‘sex estimation’ if from anthropological assessment
AW: We changed ‘sex determination’ to ‘sex estimation’.
- ‘thigh bone’ should be 'femur'
AW: We changed ‘thigh bone’ to 'femur'.
- Please clarify what bones were used, was this all femurs or a mixture? The archaeological bones in particular are not specified.
AW: We used the femur from all samples. We added the following information in the Materials and Methods part: The diaphysis of the femur of forensic and archaeological bone samples was used for analyses.
General comments:
- There is considerable male bias in the population/data used (16 female and 88 male) – please could the authors address the potential implications of this in the discussion
AW: We added the following sentence and a reference: Also, the gender imbalance between the samples most likely does not lead to a bias in the results. This observation is also demonstrated in [60], that the inter-gender variation is much bigger than the gender difference for most bone-related parameters.
- Authors frequently note the method as being non-destructive. However, the removal of bone samples from a bone/body is an invasive process and somewhat destructive to the bone. Please could this be acknowledged in the manuscript. Moreover, if the NIR could actually be used on complete bones, or even on bones still in situ (exposed but in a body) then this could also be worth noting.
AW: We agree, deleted the misleading information and added the following information: Potential applications include distinguishing between human and animal bone fragments, on complete bones, or even on bones still in situ.
- The terms ‘outside’ and ‘inside’, should be replaced with ‘external’ and ‘internal’ bone.
AW: We changed ‘outside’ and ‘inside’ with ‘external’ and ‘internal’.
- Only 5 archaeological bones were used (please discuss this limitation)
AW: We added the following information in the discussion: The spectra between the first four and the fifth class vary to such an extent in amplitude and shape, as shown in figure 4a, that the distinction between the first four and the fifth class could even be made without a machine learning approach. This indicates that the limited number of samples did not impose a limitation to the results of this study.
- The actual PMI of the archaeological bones has not been provided, except a value of ‘up to 2000 years’ is mentioned in summary – please include full sample details in the method
AW: We added a table in the text and in in the supplementary.
- Results are largely missing – please include your data or at least an exemplar
AW: We may not fully understand this question, as we are not aware of any missing results: Comparisons of external and internal sites are given in figure 3, and mean spectra of the five different age classes are provided in figure 4a. We think that the means used for further analyses represent best the data we then used for the confusion matrix.
- If known, please detail what molecules the NIR-spectrometry is/could be measuring
AW: We added the following explanation in the discussion section. An exact assignment to chemical compounds cannot be made because the incoming infrared light excites every state that matches its energy. Such states can be a single atom or molecule energy levels, vibrational or rotational states, but also vibrational or rotational states of complex molecules or lattice vibrations in general. The presented smooth spectrum, without any distinct peak leads, does not allow for the extraction of distinct molecule spectra.
- It is unclear to me what exactly is indicating the age classes? Is this different peaks? Please demonstrate with data/examples.
AW: Thank you for this important hint. In the figure 4a, the lines representing the different age are not clearly distinguishable. We changed the thickness of the lines to better differentiate the spectra of the 5 age classesThe machine learning approach allows for the extraction of intrinsic information from spectra without the necessity of knowing what information one is looking for. The neural network learns the features of the data and extracts the required information to form a decision on the age classes. The learned feature extraction capabilities of the network are encoded in the weights of the network. The neural network forms an abstract representation of the data. Those weights and the derived representations do not present themselves in terms of wavenumbers or similar terms. I.e. the neural network draws it’s information by transforming the input data in a preferable, training-based, pattern.
- Regarding section 2.5 – testing of unknown samples is mentioned, it is unclear to me if the results from this are reported. What were these used for?
AW: Indeed, section 2.5 is misleading. We therefore removed the whole section 2.5.
- The title explicitly refers to a ‘handheld’ device, however, in Figure 5, the device appears to be a small desktop device, not one that is held in a hand. I would suggest the authors reconsider the use of ’handheld’.
AW: Indeed, the combination of 8 handheld devices can be considered a small desktop device. However, each single device combined with an ANN software developed according to this study can be used as a handheld device, which can be used independently from a desk. This sentence was added to result 3.4.
With thanks and kind regards
Johannes Pallua, corresponding author
Priv.-Doz. MMag.Dr.rer.nat. Johannes Pallua MSc PhD
Univ.-Klinik für Orthopädie und Traumatologie
Anichstraße 35
A-6020 Innsbruck
Tel.: +43 50 504 80242
Mail: johannes.pallua@tirol-kliniken.at
Reviewer 2 Report
The study is very interesting and well structured.
The authors have focused on an important aspect of forensic practice with innovative instrumental approach.
The evaluation of the PMI is very difficult, the authors performed a relevant study from a scientific and technical point of view.
Author Response
Reviewer 2:
The study is very interesting and well structured.
The authors have focused on an important aspect of forensic practice with innovative instrumental approach.
The evaluation of the PMI is very difficult, the authors performed a relevant study from a scientific and technical point of view.
AW: Thank you for your review
With thanks and kind regards
Johannes Pallua, corresponding author
Priv.-Doz. MMag.Dr.rer.nat. Johannes Pallua MSc PhD
Univ.-Klinik für Orthopädie und Traumatologie
Anichstraße 35
A-6020 Innsbruck
Tel.: +43 50 504 80242
Mail: johannes.pallua@tirol-kliniken.at